# Immune Checkpoints in Recurrent Pregnancy Loss: New Insights into a Detrimental and Elusive Disorder

**DOI:** 10.3390/ijms241713071

**Published:** 2023-08-22

**Authors:** Luca Marozio, Anna Maria Nuzzo, Eugenio Gullo, Laura Moretti, Emilie M. Canuto, Annalisa Tancredi, Margherita Goia, Stefano Cosma, Alberto Revelli, Alessandro Rolfo, Chiara Benedetto

**Affiliations:** 1Department of Surgical Sciences, Obstetrics and Gynecology 1, University of Turin, Via Ventimiglia 1, 10126 Turin, Italy; eugenio.gullo@unito.it (E.G.); ecanuto@cittadellasalute.to.it (E.M.C.); atancredi@cittadellasalute.to.it (A.T.); stefano.cosma@unito.it (S.C.); chiara.benedetto@unito.it (C.B.); 2Department of Surgical Sciences, University of Turin, Via Ventimiglia 1, 10126 Turin, Italy; a.nuzzo@unito.it (A.M.N.); l.moretti@unito.it (L.M.); alessandro.rolfo@unito.it (A.R.); 3Unit of Pathology, Department of Medical Sciences, University of Turin, Via Santena 7, 10126 Turin, Italy; magoia@cittadellasalute.to.it; 4Department of Surgical Sciences, Obstetrics and Gynecology 2, University of Turin, Via Ventimiglia 1, 10126 Turin, Italy; alberto.revelli@unito.it

**Keywords:** recurrent pregnancy loss, endometrium, immune activity, placentation, T-cell activity, pregnancy, immune tolerance

## Abstract

Recurrent pregnancy loss (RPL) refers to two or more miscarriages before 20 weeks gestation. Its prevalence is 1–2%; its pathogenesis remains unexplained in more than 50% of cases, in which the cause is thought to be abnormal immune activity during placentation leading to a lack of pregnancy-induced immune tolerance. It is unknown whether immune activity is deranged in the endometrium of women with RPL. We studied the gene expression and the quantitative tissue protein levels of three immune checkpoints (CD276, which enhances cytotoxic T-cell activity, cytotoxic T-lymphocyte-associated antigen-4 [CTL-4], which reduces Th1 cytokine production, and lymphocyte activation gene-3 [LAG-3], which shows suppressive activity on Tregs and CD4+ T-cells) in endometrial samples from 27 women with unexplained RPL and in 29 women with dysfunctional uterine bleeding and previous uneventful pregnancies as controls. RNA isolation, real-time PCR, protein isolation, and ELISA were performed. CD276 gene expression and protein tissue levels were significantly lower in the endometrium of the RPL group than in the controls, whereas both CTL-4 and *LAG-3* were significantly higher. This difference suggests defective endometrial immune regulation and overactivation of immune response in women with a history of RPL, at least in relation to controls with dysfunctional uterine bleeding and previous normal reproductive history.

## 1. Introduction

A proper immunological dialogue between mother and embryo takes place during implantation and pregnancy. Immune mechanisms normally induce maternal immunological tolerance towards the trophoblast and the embryo while maintaining full immunological reactivity against all other non-self antigens; however, experimental and clinical evidence suggests that endometrial immune derangements may play a role in adverse pregnancy outcomes, such as preeclampsia, placental insufficiency, fetal growth restriction, and recurrent pregnancy loss (RPL) [1]. RPL refers to two pregnancy losses prior to 20 weeks gestation. Its prevalence is 1–2%. The risk of RPL increases with maternal age and the number of successive losses. Other RPL risk factors are parental structural chromosomal abnormalities, maternal obesity, lifestyle factors (e.g., tobacco and excessive alcohol use), uterine malformations, endocrine and metabolic disorders, antiphospholipid antibodies syndrome, and inherited thrombophilia. The comprehensive diagnostic evaluation identifies RPL pathogenesis in fewer than 50% of couples. Abnormal immune activity has been suggested as the cause in unexplained cases [2].

Moreover, the role of the endometrium is fundamental in human reproduction. It interacts with the trophoblast, thus allowing for its proper implantation, correct proliferation, and development. Throughout these events, several cell types of the innate and adaptive immune system are involved in endometrial remodeling and induction of maternal tolerance towards the embryo. Such cells develop specific characteristics when they reach the endometrium, together with an increasing number of immunoregulatory molecules (e.g., CD276, LAG3, CTLA-4).

CD276, also called B7-H3, is a member of the immune co-stimulatory molecule B7 superfamily. It was originally discovered as a co-stimulator of CD4+ and CD8+ cytotoxic T-cell proliferation and IFN-γ production. Although considered a T-cell co-inhibitor [3,4], CD276 has been noted to have contradictory effects on T cells and that it increases CD4+ and CD8+ T-cell proliferation, enhancing cytotoxic T-cell activity [5]. Cytotoxic T-lymphocyte-associated antigen-4 (CTLA-4), expressed intracellularly in regulatory T cells (Tregs), is present on the surface of Tregs, CD8+, and CD4+ T cells [6]. Its inhibitory effect is a result of the removal of its ligands CD80/CD86 from the cell surface of antigen-presenting cells (APCs), thus preventing their binding to the co-stimulatory CD28 present on T cells [7,8,9]. CTLA-4 expression at the maternal–fetal interface shows a positive correlation with decidual Th2 cytokine production and a negative correlation with decidual Th1 cytokine production, suggesting local immunosuppressive effects [10].

Lymphocyte activation gene-3 (LAG-3) is an immune checkpoint of the immunoglobulin superfamily. It enhances the immunosuppressive activity of Tregs, downregulates CD4+ T-cell activity, and has a synergic effect with CTLA-4 [11,12,13]. LAG-3 is expressed in immune cells that inhibit T-cell proliferation, activation, and homeostasis [14,15]. Furthermore, LAG-3 can downregulate CD4+ T-cell activity by binding to major histocompatibility complex II (MHC class II) [16,17].

Pregnancy is a physiological model of immunotolerance, in which the maternal immune system accepts the semi-allogeneic fetus without rejecting it while mounting an immune defense against non-self antigens. The role of CD276, LAG3, and CTLA-4 in this mechanism holds special interest, but little has been reported on their endometrial expression in women with a history of RPL. To fill this gap, with the present study, we investigated endometrial CD276, LAG3, and CTLA-4 immune checkpoints in non-pregnant women with a history of RPL.

## 2. Results

No significant differences were observed for age at hysteroscopy, body-mass index, tobacco use, and ethnicity between the RPL and the control group (Table 1). Twelve women in the RPL group had a history of 3 miscarriages and 15 had 4 or more previous miscarriages. The mean gestational age at miscarriage was 8.6 ± 0.2 weeks. As expected, immune-histochemical analysis showed positive staining for CD138, CD20, CD3, and TIA 1, thus confirming an activated inflammatory profile in the RTL group. The positive staining for estrogen and progesterone receptors in both groups confirmed normal endometrial receptivity.

There was a significant decrease in CD276 gene (Figure 1A, 0.74-fold decrease, *p* = 0.001) expression levels in the RPL endometrium (Average = 0.74, SE = 0.21) vs. the CTRL endometrium tissue samples (Average = 1, SE = 0.70). A significant decrease was also found in CD276 protein levels (Figure 1B, 0.73-fold decrease, *p* = 0.02) in the RPL endometrium (Average = 0.73, SE = 0.19) vs. the CTRL endometrium tissue samples (Average = 1, SE = 0.43). In contrast, a significant increase was observed in CTLA4 gene (Figure 2A, 2.6-fold increase, *p* = 0.001; RPL-END Average = 2.60, RPL-END SE = 2.65, CTRL-END Average = 1, CTRL-END SE = 0.87) and protein (Figure 2B, 2.4-fold increase, *p* = 0.01; RPL-END Average = 2.40, RPL-END SE = 1.52, CTRL-END Average = 1, CTRL-END SE = 0.64) expression in the RPL compared to the CTRL samples. Finally, there was a significant increase in LAG3 gene (Figure 3A, 1.4-fold increase, *p* = 0.04; RPL-END Average = 1.43, RPL-END SE = 0.89, CTRL-END Average = 1, CTRL-END SE = 0.65) and protein (Figure 3B, 1.48-fold increase, *p* = 0.04; RPL-END Average = 1.48, RPL-END SE = 0.12, CTRL-END Average = 1, CTRL-END SE = 0.35) levels in the RPL compared to the CTRL endometrium tissue samples. Even with the presence of significant differences between gene and protein expression levels in CTRL relative to RPL endometrial biopsies, we did not find any significant Pearson correlation in CD276 and LAG-3 between groups. However, we reported a significant Pearson coefficient in CTRL4 protein expression in RPL relative to the CTRL CTLA4 gene (r = 0.8; *p* = 0.02) and RPL CTLA4 gene (r = 0.71; *p* = 0.05).

## 3. Discussion

The endometrium comes into direct contact with the trophoblast, which is immunologically different and is usually considered as a semi-allograft or as a complete allograft in oocyte donation [18]. The endometrium provides an optimal endocrine/paracrine, immune, and molecular environment for correct trophoblast implantation, invasion, and development, and for full maturation of the placenta. During these processes, the endometrium undergoes unique and striking adaptive changes collectively termed decidualization. This activity involves cell reprogramming, tissue remodeling, changes in gene expression and post-translation regulation, and modification in cell signaling pathways, accompanied by local adjustments of immune cells activity. Many modifications in the endometrium and in the decidua during implantation and throughout pregnancy are mediated by immune cells activity and by diverse immunoregulatory molecules. Some data suggest that alteration in normal immune function can occur in these tissues in women with RPL.

Our observation of high positivity for CD138 in the endometrial biopsies from the RPL group is shared by Rimmer and colleagues, who immunohistochemically quantified CD138+ cells in the endometrium to assess chronic endometritis in women with the risk of RPL [19]. Endometrial CD138 positivity is often associated with abnormal inflammatory cytokine and chemokine levels; we noted an unfavorable endometrial immune status necessary for successful pregnancy in the RPL group.

CD3 T cells are among the most abundant subsets of leukocytes in the human endometrium. CD3 immunostaining positivity has been extensively investigated in women with defective endometrial receptivity and RPL; however, the results are controversial. A few studies have reported no difference in the number of CD3+ T cells in the peripheral blood and endometrium of women with recurrent miscarriage compared to healthy women [20,21], whereas we found high positivity for CD3+ cells in the endometrium tissue samples from the RPL group. Similarly, Galgani and colleagues noted an association between CD3 positivity and an altered pattern of circulating inflammatory molecules [22]. Accordingly, our hypothesis was that altered expression of endometrial plasma cells and T lymphocyte subsets, as well as an increase in circulating pro-inflammatory cytokines, would create an unfavorable endometrial environment and result in pregnancy loss.

Here, we report an aberrant expression of CD276, CTLA4, and LAG3 immune checkpoint molecules necessary for the establishment of maternal–fetal immunotolerance in the endometrium of women with a history of RPL; this finding suggests an over-activated immune maternal response in women with a history of RPL. We are aware that our control group, consisting of subjects with dysfunctional uterine bleeding and previous uneventful pregnancies, cannot be considered a healthy–fertile women’s group. However, since the controls have a normal reproductive history, they are probably more similar to healthy–fertile women than the cases with RPL. 

While aberrant CD276 expression in autoimmune diseases and cancer has been widely studied [23], its role at the maternal–fetal interface and in pregnancy-related disorders is controversial. Initially identified as a co-stimulatory factor of the immune response, recent studies have shown that it is predominantly a T-cell co-inhibitory molecule [3,24]. CD276 knockout mice showed higher differentiation of Th1-induced in response to more severe airway inflammation, more rapid development of experimental autoimmune encephalomyelitis, and higher autoantibody concentrations. Moreover, there is evidence for a correlation between CD276 expression and the intensity of Tregs infiltration in patients with esophageal squamous cell carcinoma [25]. This finding supports the hypothesis that CD276 is directly involved in fetal immune tolerance by promoting Th2 differentiation, Treg induction, T-cell suppression, and IFN-γ production [26]. In line with these data, we postulated that CD276 downregulation in the RPL group would lead to a misbalance of immunosuppressive cells/molecules and insufficient fetal immune recognition resulting in spontaneous fetal loss.

CTLA-4 is expressed on Tregs and conventional T-cell surfaces, where it acts as a brake on the activation of effector T cells [27]. Previous studies of CTLA-4 expression have shown a statistically significant relationship with the incidence of spontaneous miscarriage; however, the data are conflicting, and little is known about CTLA-4 in the endometrium or the decidua of women with a history of RPL. In line with our data, Wang and colleagues found higher CTLA-4 expression in the endometrium samples from women with RPL than in the controls [28] and a correlation with negative regulation of endometrium receptivity that contributes to RPL. CTLA-4 overexpression in RPL indicates inhibition of T-cell activation at the maternal–fetal interface, whereas the inhibitory signal in normal pregnancy is attenuated. These findings are consistent with the hypothesis that an immune response is needed to maintain pregnancy [29,30,31] and that immunologic tolerance during pregnancy is an active T-cell reaction. T-cell activation excessively suppressed by CTLA-4 suggests that neither an effective immunotolerance nor a beneficial Th2 bias immune response has been induced.

LAG-3 has been described as a marker of T-cell activation; its expression is rapidly induced in peripheral blood T and NK cells by T-cell receptors or pro-inflammatory cytokine stimulation [32,33,34]. Consistent with these initial findings, our data show increased LAG-3 expression in the endometrium tissue samples from the RPL group, with inappropriate inflammation in mid-follicular endometrial stromal cells and aberrantly elevated uNK cell density leading to abnormal angiogenesis and impaired endometrial decidualization [35]. Therefore, we may postulate that LAG-3 acts predominantly as an immune response stimulator rather than as a suppressor in the endometrium. In line with our observations, previous studies showed that the antibodies that deplete LAG-3 prolonged transplant survival in a cardiac allograft model [36], producing sustained T-cell-driven inflammation inhibition in a non-human primate delayed-type hypersensitivity challenge [37]. Beneficial effects following LAG-3 downregulation induced by monoclonal antibodies were also observed in the treatment of psoriasis in humans [38]. 

In the present study, we excluded women with histological signs of chronic endometritis, which is seldom included in most diagnostic protocols for RPL. A previous study reported chronic endometritis in 27% of women with unexplained RPL [39]. This finding is consistent with other studies that reported chronic endometritis 13–56% of women with RPL [40,41] and supports the concept that it may be a cause of RPL. However, our observation of altered endometrial immune marker levels was independent of chronic endometritis. A possible mechanism by which chronic endometritis might cause RPL is by altering normal decidualization [42]. Appropriate treatment of chronic endometritis in women with RPL may improve live birth rates [43], as well as benefit a subset of these patients. 

In conclusion, derangement of the immune system may play a role in unexplained RPL, when no other clinical causes or explanation can be found. Our data strengthen the concept that defective endometrial immune-regulation may lead to anomalies in feto-maternal tolerance and thus cause or contribute to RPL. Pregnancy is a good model for investigating immune tolerance. In addition, the study of immune tolerance in pregnancy may not only improve perinatal outcomes but also knowledge that can be transferred to many other areas of immunology. In autoimmune disorders, for example, pregnancy may induce clinical improvement and remission of symptoms of multiple sclerosis, rheumatoid arthritis, autoimmune hepatitis, and thyroiditis [44,45,46,47]. A recent study reported that immune tolerance to the first cells of the genetically foreign fetal conceptus is sustained by a dynamic and responsive immune system and not by a suppressive one, and that immune activity is similar to defense against genetically aberrant malignant cells that eventually develop into cancer [48]. Furthermore, our understanding of how the fetal allograft averts rejection may inform the design of improved antigen-specific therapies for protecting donor allografts in organ transplantation.

Future research into the role of the immune system in the pathogenesis of unexplained RPL is needed. Current treatment for RPL of supposed immune etiology is limited, empirical in most cases, and often inefficacious. In these cases, the treatment should aim to correct abnormal decidualization and dysfunction of immune cells in the endometrium and decidua. 

## 4. Materials and Methods

The study sample was 27 non-pregnant women with a history of three or more unexplained miscarriages prior to 20 weeks gestation (age range, 18–40 years) and no previous normal pregnancy. Exclusion criteria were RPL due to genetic or chromosomal causes, uterine malformations, acquired uterine disorders (fibroids, polyps, or synechiae), antiphospholipid antibodies syndrome, inherited thrombophilia, endocrine or metabolic dysfunction (hypothyroidism, diabetes, polycystic ovary syndrome), systemic diseases, autoimmune disorders (including celiac disease), and genital infections. The controls were 29 non-pregnant women with no history of RPL, infertility, or pregnancy failure (age range, 18–40 years), with at least one previous normal pregnancy within the past 10 years prior to recruitment, and with no chronic systemic diseases or genital infections. An additional criterion of exclusion was histological signs of chronic endometritis, which hysteroscopy revealed in 5 women (4 in the RPL group and 1 in the control group). 

Endometrial biopsies were obtained in sterile conditions during hysteroscopic gynecological evaluation. Hysteroscopy was a clinical step scheduled in the diagnostic protocol for RPL. The controls underwent hysteroscopy to investigate previous abnormal uterine dysfunctional bleeding. The duration of bleeding before hysteroscopy was at least 6 months (range 6–9 months). Diagnosis of fibroids, endometrial polyps, or endometrial hyperplasia was a further criterion for exclusion from the present study. Hysteroscopic biopsies were obtained in the mid-luteal phase of the cycle, between days 18 and 22, calculated from the first day of the last menstrual cycle. All subjects included had regular cycles. Endometrial biopsies were divided into two portions: one was formalin-fixed and paraffin-embedded for histological examination, and the other was immediately frozen at −80 °C for RNA and protein isolation.

The study was conducted according to the principles expressed in the Declaration of Helsinki and approved by the Institutional Ethics Committee (Prot. No 0112760). All patients were recruited at the Sant’Anna University Hospital, University of Turin (Italy), and provided written informed consent for endometrium collection and subsequent analysis. 

### 4.1. Immunohistochemistry (IHC)

Endometrial immune-phenotype characterization in the women with RPL was obtained with endometrial parallel sections (3 μm) cut, deparaffined, and incubated overnight at 4 °C with primary antibodies against CD138 (plasma cell marker), CD20 (B lymphocyte marker), CD3, and TIA1 (T-cell markers), and progesterone and estrogen receptors. The peroxidase ABC method (Vector Laboratories, Burlingame, CA, USA) was performed for 1 h at room temperature, and 3′,3′ diaminobenzidine hydrochloride (Sigma, St Louis, MO, USA) was used as the chromogen.

### 4.2. RNA Isolation and Real-Time PCR

Total RNA was isolated from endometrial biopsies using TRIzol reagent (Sigma-Aldrich, Milan, Italy) according to the manufacturer’s instructions. Genomic DNA contamination was removed by DNAse I digestion before RT-PCR. RNA was quantified using a NanoDrop microvolume spectrophotometer (ThermoFisher Scientific, Waltham, MA, USA,) and three μg of total RNA was reverse transcribed using a random hexamers approach (Fermentas Europe, St. Leon-Rot., Germany) and a RevertAid H Minus First Strand cDNA Synthesis kit (Fermentas, Cat. No k1632). Gene expression levels of *CD276* (Life Technologies, Carlsbad, CA, USA, Cat. No 4331182, seq. Hs00987207_m1), *CTLA4* (Life Technologies, Cat. No 4331182, seq. Hs00175480_m1), and *LAG3* (Life Technologies, Cat. No 4331182, seq. Hs00958444_g1) were determined by real-time PCR using specific TaqMan primers and probes following the manufacturer’s protocol (Life Technologies, Cat. No 4331182). For relative quantitation, PCR signals were compared among groups after normalization using ribosomal 18S RNA expression as the internal reference (Life Technologies, Cat. No 4333760F). Relative expression and fold change were calculated by the 2^−ΔΔCt^ as described in Livak and Schmittgen [49].

### 4.3. Protein Isolation and Enzyme-Linked ImmunoSorbent Assay (ELISA) 

Total proteins were isolated from frozen endometrial biopsies using PBS (phosphate-buffered saline) (1X, pH 7.4), and protein concentrations were then determined using Bradford protein assay (Biorad Laboratories, Milan, Italy). Once normalized the total protein concentrations of each sample to the lowest one, the quantitative measurement of CD276 (Cusabio, Houston, TX, USA, Catalog Number CSB-E14285h), LAG3 (Cusabio, CSB-EL012719HU) and CTLA-4 (Cusabio, Catalog Number CSB-E09171h) endometrial levels were determined using commercially available competitive ELISA kits according to the manufacturer’s instructions. Briefly, the samples were incubated in 96-well plates pre-coated with a capture antibody directed against CD276, LAG3, or CTLA-4 for 2 h. The wells were then washed three times and incubated with a secondary antibody against CD276, LAG3, or CTLA-4 conjugated with horseradish peroxidase. The plates were then washed three times, a substrate solution containing H_2_O_2_ and tetramethylbenzidine was added, and optical density was determined at 450 nm. All assays were conducted in duplicate. The protein levels were calculated using a standard curve derived from known concentrations of the respective recombinant proteins.

### 4.4. Statistical Analysis

Data are presented as mean ± standard error of the mean (SE). Comparison between groups was performed by analysis of variance. Bonferroni’s test was used for post hoc comparison between groups. Categorical variables are presented as frequencies (percentages), and the comparison between groups was performed using the chi-square test. In CTRL and RPL groups, we also investigated whether there is a correlation between gene and protein CD276, LAG3, or CTLA-4 expressions by calculating the Pearson correlation coefficient. Statistical analysis was performed using IBM SPSS for Windows Version 27 (IBM Corp, Armonk, NY, USA). Statistical significance was accepted at *p* < 0.05. 

## Figures and Tables

**Figure 1 ijms-24-13071-f001:**
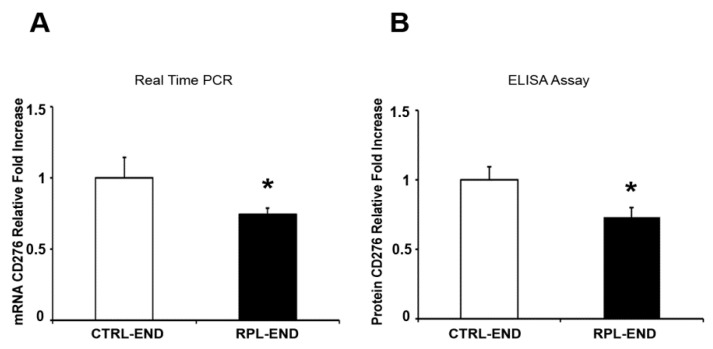
CD276 gene (**A**) and protein (**B**) expression levels in endometrial biopsies. (**A**) Fold change of CD276 mRNA in endometrium tissue samples from the RPL and the control group. (**B**) Fold change of CD276 protein in endometrium tissue samples from the RPL and the control group. Data are presented as mean ± SE. Significance was calculated using Student’s *t*-test (* *p* < 0.05). RPL denotes recurrent pregnancy loss; CTRL-END: endometrial tissue from the control group; RPL-END: endometrial tissue from the RPL group.

**Figure 2 ijms-24-13071-f002:**
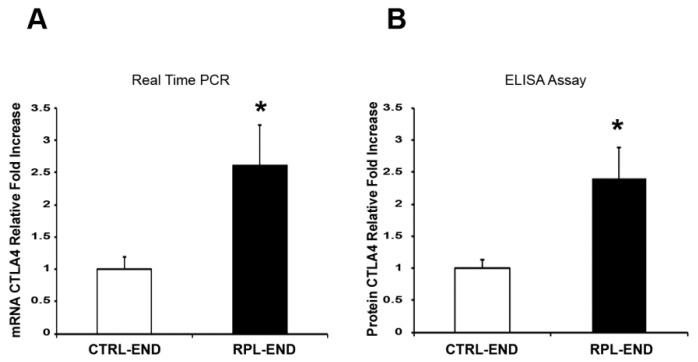
CTLA4 gene (**A**) and protein (**B**) expression levels in endometrial biopsies. (**A**) Fold change of CTLA4 mRNA in endometrium tissue samples from the RPL and the control group. (**B**) Fold change of CTLA4 protein in endometrium tissue samples from the RPL and the control group. Data are presented as mean ± SE. Significance was calculated using Student’s *t*-test (* *p* < 0.05). RPL denotes recurrent pregnancy loss; CTRL-END: endometrial tissue from the control group; RPL-END endometrial tissue from the RPL group.

**Figure 3 ijms-24-13071-f003:**
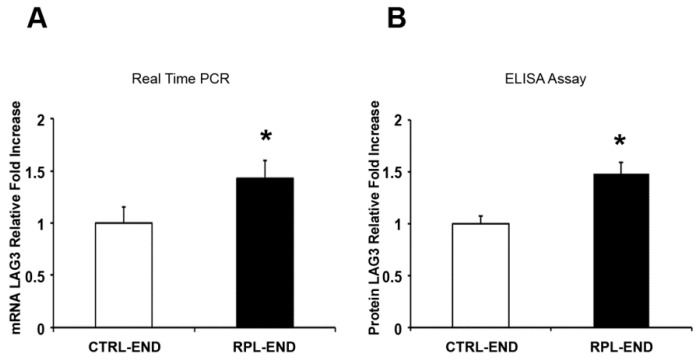
LAG3 gene (**A**) and protein (**B**) expression levels in endometrial biopsies. (**A**) Fold change of LAG3 mRNA in endometrium tissue samples from the RPL and the control group. (**B**) Fold change of LAG3 protein in endometrium tissue samples from the RPL and the control group. Data are presented as mean ± SE. Significance was calculated using Student’s *t*-test (* *p* < 0.05). RPL denotes recurrent pregnancy loss; CTRL-END: endometrial tissue from the control group; RPL-END endometrial tissue from the RPL group.

**Table 1 ijms-24-13071-t001:** Sample demographic characteristics. RPL denotes recurrent pregnancy loss.

Characteristic	RPL Group (N = 27)	Control Group (N = 29)	*p*-Value
Age at hysteroscopy—years, mean ± SE	36.4 ± 0.78	38.1 ± 0.74	NS
Body-mass index (kg/m^2^)—mean ± SE	23.4 ± 0.24	24.2 ± 0.27	NS
Tobacco use—no. (%)	2 (7.4)	1 (3.7)	NS
Caucasian ethnicity—no. (%)	23 (85.2)	24 (82.7)	NS

## Data Availability

No new data were created or analyzed in this study. Data sharing is not applicable to this article.

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
