# Peer review of "Immune Checkpoints in Recurrent Pregnancy Loss: New Insights into a Detrimental and Elusive Disorder"

_ijms, 2023, doi:10.3390/ijms241713071_

Round 1

Reviewer 1 Report

The manuscript is missing important information

In Materials and methods authors write “Endometrial biopsies were obtained in sterile conditions during hysteroscopic gynecological evaluation".
-On what day of the menstrual cycle was the biopsy performed? How was this day determined? Did all patients and control women have a regular cycle?
  It’s very important information.

In Materials and methods authors write "Total proteins were isolated from frozen endometrial biopsies using PBS (phosphate buffered saline) (1X, pH 7.4)".  and  “Endometrial biopsies were divided into two portions one was formalin fixed and paraffin embedded for histological examination and the other was immediately frozen at  -80 °C for RNA and protein isolation. 

- “Endometrial biopsies portions”-  Is how much in International System of Units?
- Endometrial samples (1g) were unlikely to dissolve completely in 1ml of PBS. Was the total protein concentration that was dissolved from the frozen samples determined? If yes, this information should be added and the concentration of CD276, LAG3 or CTLA-4 re-calculated for total protein (per mg). If the total protein was not determined- how can we know that the difference in concentration is not the result of a difference in the protein isolation efficiency?

In Materials and methods authors write “The controls underwent hysteroscopy for investigating previous abnormal uterine dysfunctional bleeding. Diagnosis of fibroids, endometrial polyps or endometrial hyperplasia was a further criterion for exclusion from the present study”.

-So control group it’s a women’s with idiopathic abnormal uterine dysfunctional bleeding? That previously had at least 1 successful pregnancy.

-How long ago have they bleeding?

- How long ago have they been pregnant or given birth??

-Did they have a history of infertility and pregnancy failures?

-All this information should be added to Table 1.

And most importantly, the control group is not healthy- fertile women But women with idiopathic abnormal uterine dysfunctional bleeding. Thus, questions arise regarding the conclusions made in the research.

In Conclusion authors write "This difference suggests defective endometrial immune-regulation as a cause of local overactivation of immune response in women with a history of RPL, potentially resulting in an unfavorable endometrial environment and pregnancy loss"

-And why not =This difference suggests defective endometrial immune-regulation as a cause of local HYPO-activation of immune response in women with a history of uterine dysfunctional bleeding?

This should be explained. Or, in the text of the article and the abstract, it should be added that the detected difference was found between two clinical groups and which one is more similar to healthy-fertile women is unknown.

-In Fig 1 2 and 3 authors show difference in average levels CD276, LAG3 and CTLA-4 between groups.  The individual levels and their distribution in groups should also be added.
-The correlations between CD276, LAG3 and CTLA-4 mRNA and proteins concentration should also be added.

Author Response

We thank the reviewer for the comments and suggestions. The point-by-point responses to the reviewer comments are the following:

1) Hysteroscopic biopsies were performed in the mid-luteal phase of the cycle, between the 18th and 22nd day calculated from the first day of the last menstrual cycle. All subjects included had regular cycles. This information has been added in the Section “Materials and Methods”.

2) As reported in paragraph 4 “Materials and Methods”, endometrial biopsies were obtained during hysteroscopic gynecological evaluation and no standards biopsies size are possible. However, once obtained endometrial biopsies, they were divided into two equal portions: one was formalin fixed and paraffin embedded for histological examination and the other was immediately frozen at -80 °C for RNA and protein isolation. For IHC analyses, endometrial parallel sections of 3 μm were cut (Paragraph 4.1). Therefore the section dimensions were comparable between samples bypassing the possible differences in biopsies dimensions. Instead, in order to overcome the possible heterogeneity of endometrial portion dimensions for RNA and protein isolations, it is important to underline that total RNA and protein were extracted but always the same quantity of RNA (3 μg) or protein were used to analyze and compare CD276, LAG3 or CTLA-4 expressions between samples following manufacturers’ instructions. Gene expression levels of CD276, LAG3 or CTLA-4 were  normalized for housekeeping gene (18s) expression as mentioned in the text (page 8, line 301-303). For ELISA assay, we determined the total protein concentration of our samples using Bradford assay (Biorad Laboratories, Milan, Italy)  and then the concentration of each samples was normalized to the lowest one. Therefore, an equivalent amount of proteins in each sample was used to analyze the difference in concentration of  CD276, LAG3 and CTLA-4 between the two groups.  The text was modified accordingly (“Materials and Methods”, par. 4.3).

3) Frozen endometrial biopsies were homogenated in PBS 1X (that represents the concentration) to isolate total proteins. As mentioned above, we determined the total protein concentration of our samples using Bradford assay (Biorad Laboratories, Milan, Italy)  and then the concentration of each samples were normalized to the lowest one. Therefore, and initial equivalent amount of proteins in each sample were used to analyze the difference in concentration of  CD276, LAG3 and CTLA-4 between the two groups.  The text was modified accordingly (“Materials and Methods”, par. 4.3).

4) In the control group, the duration of bleeding before hysteroscopy was at least six months (range 6-9 months), previous normal pregnancies were within the past 10 years prior to recruitment in the study, and the subjects had not previous history of infertility or pregnancy failure.

We do not consider these data suitable to be added in Table 1, since they are related only to controls. The information has been added in the Section “Materials and Methods”.

5) The conclusion in the Abstract and in the Discussion have been changed according to the reviewer’s suggestion, pointing out that our control group cannot be considered a group of fully healthy-fertile women.

6) Following Reviewer 1 suggestion, we added individual values of average and standard error of the mean (SE) that represent the standard deviation of its sampling distribution or an estimate of that standard deviation in the text (Section “Results”)

7) In order to investigate whether there is a correlation between CD276, LAG3 and CTLA-4 mRNA and proteins levels, we calculated Pearson correlation coefficient. Even the presence of significant differences between gene and protein expression levels in CTRL relative to RPL endometrial biopsies, we didn’t find any significant Pearson correlation in CD276 and LAG-3 between groups. However, we reported a significant Pearson’ coefficient in CTRL4 protein expression in RPL relative to CTRL CTLA4 gene (r=0,8; p=0,02) and RPL CTLA4 gene (r=0,71; p=0,05). The text was modified accordingly (Section “Materials and Methods”, par 4.4).

All the revisions are written in red.

Reviewer 2 Report

Dear authors, 

Defective endometrial T cell immune response, as a main cause of local story of RPL, could be analysed simultaneously with effects of TGF betta and other grouth factors related to development of RPL and fibroid uterine tumours, potentially also resulting from  unfavorable uterine cellular snd mollecular processes. Best regards, Reviewer

update

I would like to point out that the molecular biology is very well developped in Italy.

The great contributions of our authors will give further possibility to involve immunology methods not only in oncology, infectious diseases, etc., but first time in the treatment of PRL of immune etiology.

This discovery of authors is a sufficient reason the paper to be accepted by Editors and successfully edited.

Resuls on the role of immune system in the pathogenesis of "unexplained RPL" are not yet clarified in medical literature and clinical practice, thus further studies and scientific projects, are needed.

Best regards and nice wishes for future successes,

Second update

References are completed and well related to the text.

Text is real, and every "improvement" will be "faulse-postive".

Figures and explanations are well coordinated to content of the article.

I have not more comments and could only recommend to the  Editorial

Staff to accept and edit the prestigious article.

Best regards,

Author Response

We thank the reviewer for the kind comment to the manuscript.

Round 2

Reviewer 1 Report

I am satisfied with the changes in the manuscript and the authors' responses.